# Lessons Learned from Somatic Cell Nuclear Transfer

**DOI:** 10.3390/ijms21072314

**Published:** 2020-03-27

**Authors:** Chantel Gouveia, Carin Huyser, Dieter Egli, Michael S. Pepper

**Affiliations:** 1Institute for Cellular and Molecular Medicine, Department of Immunology and South African Medical Research Council (SAMRC) Extramural Unit for Stem Cell Research and Therapy, Faculty of Health Sciences, University of Pretoria, Pretoria 0002, South Africa; michael.pepper@up.ac.za; 2Department of Obstetrics and Gynaecology, Reproductive Biology Laboratory, University of Pretoria, Steve Biko Academic Hospital, Pretoria 0002, South Africa; carin.huyser@up.ac.za; 3Department of Obstetrics and Gynecology, Columbia University Medical Center, New York, NY 10027, USA; de2220@cumc.columbia.edu

**Keywords:** SCNT, oocyte, enucleation, somatic cell, nuclear transfer, cloning, epigenetic reprogramming, nuclear reprogramming, ESC

## Abstract

Somatic cell nuclear transfer (SCNT) has been an area of interest in the field of stem cell research and regenerative medicine for the past 20 years. The main biological goal of SCNT is to reverse the differentiated state of a somatic cell, for the purpose of creating blastocysts from which embryonic stem cells (ESCs) can be derived for therapeutic cloning, or for the purpose of reproductive cloning. However, the consensus is that the low efficiency in creating normal viable offspring in animals by SCNT (1–5%) and the high number of abnormalities seen in these cloned animals is due to epigenetic reprogramming failure. In this review we provide an overview of the current literature on SCNT, focusing on protocol development, which includes early SCNT protocol deficiencies and optimizations along with donor cell type and cell cycle synchrony; epigenetic reprogramming in SCNT; current protocol optimizations such as nuclear reprogramming strategies that can be applied to improve epigenetic reprogramming by SCNT; applications of SCNT; the ethical and legal implications of SCNT in humans; and specific lessons learned for establishing an optimized SCNT protocol using a mouse model.

## 1. Introduction

Somatic cell nuclear transfer (SCNT) has been an area of interest in the field of stem cell research and regenerative medicine for the past 20 years. The first step in SCNT is called enucleation and can be described as the removal of the haploid (1n) chromosomes comprised of the meiotic spindle complex from a metaphase II (MII) stage oocyte [1,2]. Enucleation is then followed by the transfer and fusion of a diploid (2n) somatic cell (obtained from a suitable donor) into an enucleated oocyte, the latter of which is known as a cytoplast [2,3]. The manipulated oocyte is then artificially activated by means of either electric pulses or chemical stimulation, which induces subsequent development of the embryo [2] (Figure 1). Somatic cell nuclear transfer is used primarily for generating cells and tissues that are immunocompatible with the somatic cell donor, a concept known as therapeutic cloning. Originally suggested in 1999, the production of patient-specific or genetically identical embryonic stem cells (ESCs) for research and therapeutic purposes emphasizes the potential of SCNT as a technique with unique applications [4].

In 1938, the initial concept of nuclear transfer was suggested by the “father of cloning”, Hans Spemann, who proposed that the nuclear genome of embryonic or somatic cells could be reprocessed and evaluated for the possibility of maintaining full-term development [5]. The first successful investigation of this concept was performed in amphibians in 1952, by the transfer of embryonic blastomere nuclei into enucleated eggs [6]. The first successful demonstration of nuclear transplantation to produce cloned frogs was achieved in 1962 and resulted in viable offspring [7]. This represents the first case to report the reprogramming of a somatic cell to a totipotent state via an enucleated egg. Due to technical and biological limitations related to oocyte manipulation, not until the early 1980s was nuclear transplantation reported in mammals. A breakthrough occurred in 1996 when Dolly the sheep was the first progeny to be created by SCNT using an adult somatic cell [8].

These milestones in nuclear transfer are representative of *reproductive cloning*, which can be described as the transfer of a cloned embryo created by SCNT into the uterus of a surrogate mother, to ultimately achieve full-term development of the cloned offspring. The concept of *therapeutic cloning* was initially demonstrated in the mouse [9], with the aim of creating and harvesting stem cells that could potentially be used to treat diseases. Subsequently, using similar SCNT techniques, many species including cattle [10,11,12,13,14], mouse [15,16,17,18,19,20], pig [21,22,23,24,25,26,27], rabbit [28,29], rhesus macaque [30,31], and several more have been cloned successfully, producing viable offspring or ESCs for the purpose of reproductive or therapeutic cloning, respectively [32]. Despite previous achievements, SCNT remains an inefficient process; many abnormalities are seen in cloned animals and the overall efficiency of creating normal viable offspring in animals by SCNT varies, ranging between 5% and 10% [33]. Blastocyst development in human oocytes after SCNT also varies; however, the realistic expected rate is 10% [34,35,36,37].

In this review, we begin by providing a brief overview of the abnormalities found in cloned animals, followed in more detail by SCNT protocol development, epigenetic reprogramming, applications, and the ethical and legal implications of SCNT in humans. All illustrative images were created by C.G. using Microsoft Word 2016.

## 2. Abnormalities in Cloned Animals

Reproductive cloning by SCNT with any donor cell type results in losses during pre- and post-implantation, as well as throughout pre- and post-natal development [33]. The first phenotype of clones is cell cycle arrest. The first defect in clones is genome instability, even before transcriptional abnormalities [38]. This shows that epigenetic processes involved in the differentiated state not only affect transcription, but also DNA replication. During development, cell-type-specific limitations in proliferation are an important component of cell differentiation.

The barriers to reprogramming are genome instability first, and, second, transcriptional reprogramming. The first barrier is a requirement for the second. The developmental defects discussed below are all later in development. It is difficult to determine cause and consequence that late in development, as a primary defect leads to secondary consequences. Miscarriage and fetal mortality rates are high and frequently observed as a result of developmental defects in live clones produced from many species, and the latter has been attributed to incomplete reprogramming of the somatic nuclei by SCNT [39]. Insufficient remodelling and reprogramming of the nucleus results in abnormal gene expression, subsequently contributing to abnormal placental and fetal development [40]. The latter has been called large offspring syndrome which is known for various phenotypes during pre- and post-natal development. During gestation, phenotypes such as hydroallantois, reduced mammary development and extended gestation have been observed [41]. Phenotypes noted at birth include large birth weight, abnormal organ size, motor control loss, enlarged tongue, and the development of respiratory problems as well as a weakened immune response in young clones soon after birth [42,43,44,45,46]. Obesity is an additional phenotype observed in adult clones [47].

However, species-specific differences do exist. At birth, bovine clones are more susceptible to obesity, whereas pig clones are underweight and have underdeveloped placentas [41,48]. Murine clones on the other hand have been associated with underdeveloped placentas in the early stages of gestation [49,50], but from the midpoint of gestation to birth there has been an association with placental hyperplasia [40,51,52]. In mice, abnormal epigenetic modifications including aberrant DNA methylation and histone modifications have been revealed in cloned embryos [53,54,55]. Furthermore, in addition to abnormal placentas [40,51], several abnormalities have been found in full-term murine offspring that have led to early death due to respiratory failure or other deformities [56,57], obesity [47], liver necrosis, tumours and pneumonia [58].

Several factors have contributed to the low efficiency of SCNT including invasive micromanipulation; oocyte incompetence and variation in developmental efficiency; in vitro culture inconsistencies; and early protocol deficiencies.

## 3. Early SCNT Protocol Deficiencies and Optimizations

The procedure of SCNT involves the removal of the meiotic spindle complex of the oocyte. As opposed to interphase nuclei, meiotic spindles are not visible in the MII stage human oocyte using conventional light microscopy [32]. Originally, enucleation protocols identified the spindles by staining the chromosomes with fluorochromes followed by exposure to ultraviolet (UV) light (Figure 2). However, exposure of oocytes to UV light is detrimental and compromises the development of SCNT embryos [32].

Following the undesirable consequences of the initial SCNT protocol, technical improvements were introduced into the standard protocol and this led to enhanced procedural efficiency in various species. One of the first technical modifications was the introduction of a non-invasive spindle imaging system in the form of polarized light birefringence, which has dramatically improved blastocyst development [59]. In monkeys, this single modification to the SCNT procedure significantly improved the blastocyst formation rate from 1% to 16% [60]. Contrary to in most mammalian species, including humans, in mice (e.g., the BDF1 strain) the meiotic spindle complex containing the chromosomes in the metaphase stage oocyte is easily visible as a translucent region under a Hoffman modulation contrast microscope (Figure 3) [3,9,61]. Based on the latter, SCNT experiments in mice do not require an identifying tool, which avoids the damage incurred by UV light or the need to purchase expensive equipment for polarized light birefringence [62]. Furthermore, oocyte lysis during enucleation was a procedural issue; however, this has been addressed by thinning and creating a hole in the zona pellucida with a laser system [63].

Transferring and fusion of the donor somatic cell nucleus to the cytoplast is another possible detrimental step. Fusion was regularly performed by whole cell electroporation, which was reported to prematurely trigger cytoplast activation and extrusion of the second polar body, followed by the continuation of meiosis in MII spindles [64]. An adjustment was made to bypass the adverse effect of premature activation caused by electrofusion through testing a membrane fusion approach using an inactivated Sendai virus, also known as hemagglutinating virus of Japan (HVJ) extract, between the donor somatic cell and the cytoplast [64]. The viral envelope is made up of fusogenic proteins that promote fusion (Figure 4) [63] and prevent the degradation of the maturation-promoting factor (MPF) that is required for successful reprogramming [32].

A study in which monkey oocytes were exposed to caffeine, a protein phosphatase inhibitor, also reported effective protection of the cytoplast from premature activation, as well as improved development of SCNT embryos [65]. Another study investigated the effects of caffeine on human oocytes during spindle enucleation and fusion and reported an enhanced blastocyst development rate and good blastocyst quality, characterized by noticeable and prominent inner cell masses (ICM), comparable to embryos produced by in vitro fertilization (IVF) [35].

Another downfall of the SCNT protocol yet to be corrected is the cell-cycle stage incompatibility between the donor somatic cell nucleus and the cytoplast, which may cause irregular DNA replication and consequently aneuploidies [66]. The oocyte cytoplasm is paused at the metaphase stage, while the somatic cell nucleus is transferred at the G0/G1 (interphase) stage of the cell cycle [67]. Ideally, the somatic cell nucleus should be synchronized and transferred at the mitotic phase of the cell cycle. This can be achieved by exposing donor somatic cells to compounds (e.g., nocodazole) that hamper microtubule polymerisation [32]. However, even minimal exposure to these drugs can be toxic or cause damage and in most cases is detrimental to normal development [68]. Additionally, oocyte activation is an essential step for the continuation and completion of meiosis, which is naturally triggered upon the entry of sperm during fertilization. The zygote relies on activation, which triggers reprogramming and metabolic activity in the oocyte cytoplasm, which in turn is required to maintain subsequent development [69]. In SCNT, artificial induction of the activation stimulus is needed since natural fertilization is sidestepped. As early as the 1980s, artificial chemical induction treatments were developed to mimic the biochemical processes activated by natural sperm stimulation [69]. Use of this treatment, however, does not always ensure complete activation. To optimize the activation protocol, cytoplasts are exposed to kinase or protein synthesis inhibitors in addition to induction of a calcium influx [32,70]. Interestingly, in human SCNT oocytes fused by the HVJ extract and activated via standard treatment, subsequent development to the blastocyst stage failed [35]. Therefore, as an additional activation stimulant and not as a cell fusion promoter, an electric pulse has previously been used to support efficient activation and reprogramming of the cytoplast after human SCNT [35].

## 4. Donor Cell Type and Cell Cycle Synchrony

In 1997, the first cloned mouse was born following the transfer of an adult cumulus cell nucleus using a novel single-step SCNT technique known as piezoelectric nuclear transfer [9]. Piezoelectric nuclear transfer offers direct nuclear injection with minimal transfer of cytoplasm from the donor cell, also bypassing the need for fusion. This is achieved by exposing the piezoelectric material to an electric field, which induces very powerful and accurate directional movement of the pipette tip to cut the zona during zona drilling and to penetrate the oolemma during nuclear transfer. To date, the somatic cell nuclear donors that are routinely used for the production of cloned mice include cumulus cells [9], tail-tip fibroblast cells [51], fetal fibroblast cells [71], immature Sertoli cells [72], and ESCs [57]. The success rate of producing cloned mice is generally higher when ESC nuclei are used as opposed to somatic cell nuclei [57,73]. This suggests that the efficiency of cloning may increase if donor cells in an undifferentiated state are used. Several reports of full-term development following SCNT with undifferentiated donor cell types, such as neuronal [74], hematopoietic [75], mesenchymal [76] and keratinocyte stem cells [77], have shown equivalent or lower efficiency rates compared to differentiated somatic cell nuclei data. One group reported that differentiated donor cells were more efficient than adult stem cells for cloning by SCNT [78]. In a recent study that cloned Macaque monkeys by SCNT using adult cumulus cells and fetal fibroblasts, SCNT efficiency using adult cumulus donor cells was lower than when fetal fibroblasts were used as donor cells [31]. This may be the result of the less efficient reprogramming of adult nuclei as opposed to fetal nuclei, or it could be attributed to a difference in the type of somatic donor cell used [31].

According to a study that compared the cloning efficiency of cattle using ovarian cumulus, mammary epithelial, and skin fibroblast donor cells, the type of donor cell can significantly affect embryo development [39]; in terms of in vitro and full-term development, cumulus donor cells were the most effective. The results suggest that cumulus cell DNA may be reprogrammed more effectively after SCNT. An earlier study in mice also reported an increase in live birth rate from cumulus donor cells when compared to Sertoli and neuronal donor cells [9]. Numerous types of somatic cell donors have been investigated; the agreement from several laboratories is that cumulus donor cells achieve the highest cloning efficiency with the lowest number of abnormalities in cloned animals [9,39,79,80]. Additionally, more than 80% of cumulus cells are arrested at the G0/G1 phase of the cell cycle, and thus are suitable donor cells that can be used without any selection criteria and do not require further in vitro culture to verify cell cycle synchronization [61,81]. See Table 1 for success rates in mice using various donor cell types.

Synchronization of the cell cycle between the recipient oocyte and the donor cell nucleus in SCNT is important to ensure successful epigenetic reprogramming and ultimately full-term development [8]. After the birth of Dolly, it was proposed that the donor cell should be in the quiescent (G0) phase of the cell cycle and to avoid the S phase during SCNT [8]. Several other studies have proposed that successful cloning can be achieved with donor cells in the G0, G1, G2, and M phases of the cell cycle [57,86,87]. In SCNT, the introduction of G0, G1, and M phase donor cells into MII stage oocytes is performed routinely, preventing both DNA damage and unplanned DNA replication of the donor cell. Compatibility between the donor cell cycle stage and elevated MPF activity in MII oocytes is recommended. In MII stage oocytes, elevated MPF activity promotes the donor cell nucleus to undergo nuclear envelope breakdown (NEBD) and premature chromosome condensation (PCC) [67,88]. Nuclear reprogramming in SCNT is believed to be promoted by PCC. Although the exact mechanism has not been identified, it has been proposed that NEBD and PCC may facilitate the release of somatic factors that are bound to chromatin. As a result, donor cell chromatin is more accessible to oocyte factors involved in reprogramming and DNA synthesis [89]. The remodelling and/or reprogramming of a somatic cell nucleus in SCNT are believed to be accelerated by high levels of MPF and mitogen-activated protein kinase (MAPK) activity; however, a recent study found that both MPF and MAPK activity are not necessary for the initial step in nuclear reprogramming and/or remodelling of the chromatin [90].

The generally accepted cause of the low efficiency of SCNT is abnormal gene expression due to failure of epigenetic reprogramming of the donor somatic cell nucleus by the oocyte [33]. The transferred somatic cell nucleus is expected to undergo a sequence of epigenetic changes caused by factors within the cytoplasm of the cytoplast. Ideally, this implies the complete removal of the “somatic donor cell memory” followed by new zygotic chromatin being established [32].

## 5. Epigenetic Reprogramming in SCNT

Epigenetics can be defined as the study of phenotypic deviations that occur in cells controlled by gene expression and the modifications made to the chromatin structure, which may switch genes on or off, without altering the genotype of the cell [33,41,91]. Numerous reprogramming events occur during cell differentiation, and SCNT is one of the most effective techniques for studying this phenomenon. Cloning with somatic cell nuclei has shown that epigenetic modifications within a differentiated genome can be altered to the totipotent state [7]. However, despite the fact that nearly 20 years have passed since the first mammal was successfully cloned from an adult somatic cell [8], the complete reprogramming of a differentiated somatic cell nucleus remains inefficient and the mechanisms by which this phenomenon is achieved are not yet fully understood.

To recognize the essential mechanisms involved in embryo development following SCNT, gene expression during normal development and regulation must be discussed. Chromatin architecture is very complex and plays a vital role in regulating gene expression. Changes in chromatin structure, and ultimately patterns of gene expression, are modulated by DNA methylation, histone subunits and the composition of nuclear lamins, and are followed by histone post-translational modifications including acetylation, phosphorylation, and methylation. Nuclear composition alters drastically during embryogenesis as well as during the specialization of nuclei in specific tissues. To ensure successful SCNT, the pattern of epigenetic modifications in the differentiated nucleus of the donor cell should undergo remodelling to become like the pattern present in the nucleus of a fertilized oocyte. In addition, the cytoplasm of the arrested MII oocyte should assist the remodelling process. Nuclear remodelling is defined as a change in chromatin structure and is known to alter the pattern of genes that are to be transcribed, which is known as nuclear reprogramming. The difference between nuclear remodelling and nuclear reprogramming is emphasised to avoid misperception, as remodelling refers to the structural rearrangement of the DNA, while reprogramming is the consequence of those physical changes [41,92].

### 5.1. Nuclear Remodelling and Reprogramming in Embryogenesis

During SCNT, the nucleus undergoes structural modifications, which are better understood by describing the general structure of the somatic cell nucleus, as well as that of the pronuclei in a zygote. The pronuclei are surrounded by a unique environment where very little to no transcription occurs within the zygotic cytoplasm, and where factors are localized that would direct the first few cell divisions after fertilization [41]. Embryonic genome activation eventually kicks in once the embryo begins producing enough ribonucleic acid (RNA), at which point transcription can begin [41]. This occurs at species-specific cell stages, for example at the two-cell stage in mice [93] and during the four-cell stage in humans [94]. At this stage, true control over embryo development is maintained by the developing embryo’s own nuclei. As the embryo passes through each developmental stage, protein associations with the nuclei change [95]. For example, when the ICM and the trophectoderm (TE) (which are the first two distinguished cell types) are formed, a specific set of genes is transcribed along with specific proteins associated with the nucleus for each different cell type [96]. Subsequently, tissue formation and specialization occur, each with their own unique nuclear structure and set of genes that are transcribed [41].

#### 5.1.1. DNA Methylation

DNA methylation and histone post-translational modifications occur after SCNT. This includes methylation, acetylation, phosphorylation and ubiquitination. In mice, DNA methylation and histone modifications have been well described during normal embryo development. However, since several species-specific differences exist, one should be mindful that patterns of development in one species do not necessarily replicate those in another. Nonetheless, the arrangement of local chromatin is altered by DNA methylation, which is generally associated with the inhibition of transcription [10]. Patterns of DNA methylation differ between early mammalian embryos and maternal and paternal genomes [97]. At fertilization, high levels of DNA methylation are present in the DNA of sperm and oocytes. During preimplantation development in the mouse, total DNA demethylation occurs as the paternal genome undergoes active demethylation after fertilization, which causes the decondensation of sperm DNA and establishment of the paternal pronucleus [98]. In addition, and contrary to the paternal genome, the maternal genome undergoes passive demethylation during the first few cell divisions [98,99]. At the approximate time of ICM and TE specialization, new patterns of DNA methylation are regulated by DNA methyltransferases, followed by lineage-specific methylation as cell specialization is determined [41].

#### 5.1.2. Histone Modifications

DNA compaction is achieved through interaction with histone proteins. Acetylation and methylation are generally the most extensively investigated histone modifications. Histone acetylation reduces the association between the histone and tightly packed heterochromatin, which usually results in newly accessible euchromatin which can undergo active transcription. Transcriptional activation or repression may be the result of histone methylation, depending on the histone residue altered [100]. In conjunction with DNA methylation, histone modifications are also changed during normal embryogenesis. Directly after fertilization in mice, the histone modification patterns of the paternal and maternal genomes are irregular. Paternal pronuclei histone H4 is hyperacetylated in comparison to that of maternal pronuclei [101]. In contrast, maternal pronuclei contain high levels of other histone residues which are not present in paternal pronuclei. At the blastocyst stage, deviation in the histone modification profiles of the ICM and TE are noted [24]. Histone modification reprogramming is more complicated than that of DNA methylation; however similar to DNA methylation, stage-specific and cell type-specific changes do occur [41,91].

#### 5.1.3. Associations of Epigenetic Events

The array of epigenetic modifications and the control of transcription during normal embryo development are complex, with optimal epigenetic regulation being reliant on the interaction between DNA methylation and histone modifications [102]. An inverse relationship exists between histone acetylation at certain sites of DNA and the resultant methylation of surrounding chromatin. Transcription is affected by several mutually supportive interactions between DNA methylation and histone modifications (Figure 5) [103]. An example of the multiple levels of epigenetic regulation in embryogenesis is the remodelling of the paternal chromatin that occurs after fertilization until the first cell division. Sperm DNA is highly compacted due to its interaction with protamine. After fertilization, protamine is removed and replaced with acetylated histones to help maintain the newly folded DNA in an open, loosely packed conformation [104]. To prepare the DNA for transcription, reprogramming of the genome is performed through the combination of histone modifications, progressive DNA demethylation, loss of oocyte-specific histones, and the acquisition of non-histone proteins [41,92,104].

### 5.2. Nuclear Remodelling and Reprogramming in SCNT Embryos

Once a somatic cell nucleus is introduced into an oocyte, several sequential events must occur to ensure successful reprogramming. The structure of the chromatin within the nucleus is remodelled, consequently erasing the differentiated epigenetic markers of the somatic cell. This is completed by reprogramming the developmental gene expression pattern to one that mimics that of a normally fertilized oocyte. Following appropriate activation, the reconstructed SCNT embryo undergoes the equivalent developmental sequence and subsequent patterns of embryonic gene expression as observed in a normal zygote [41]. The transcriptional silencing of the somatic nucleus requires structural remodelling. This includes nuclear membrane breakdown, chromatin condensation, spindle assembly, the release of somatic cell-specific proteins from the nucleus into the ooplasm, the acquisition of certain ooplasm-specific proteins from the ooplasm by the transferred nucleus, as well as the establishment of a structure comparable to a pronucleus after activation [105].

Well-known examples of protein exchange after SCNT in mice include the histone variants H1FOO and MacroH2A. Histone subunits are linked together by histone H1, which makes up the nucleosome. An oocyte-specific alternative of histone H1 exists, namely H1FOO, which quickly replaces histone H1 when a somatic cell nucleus is introduced into the ooplasm [106]. MacroH2A is present in somatic cells and absent from the nuclei of fertilized oocytes until the first few cell divisions. After SCNT, MacroH2A is eliminated from the chromatin and broken down. At the morula stage of embryo development, MacroH2A is then re-established and accumulates in the chromatin, as in normally fertilized oocytes [107].

For effective reprogramming, it is assumed that the somatic cell pattern of epigenetic modifications must be reversed prior to embryonic genome activation [108], at the two-cell stage in mice [93] and the four-cell stage in humans [94]. Incomplete epigenetic remodelling and aberrant patterns of DNA methylation or histone acetylation in SCNT embryos have been identified in numerous studies, and all contribute to the inefficiency of SCNT [15,53,109]. Following SCNT, the somatic cell genome does not react to the ooplasmic activity of dynamic demethylation, therefore SCNT embryos have increased DNA methylation levels as opposed to normal embryos [110]. Rapid deacetylation of histones is another consequence of SCNT, as well as abnormal patterns of histone methylation in SCNT embryos [24,111].

### 5.3. Improving SCNT with Chromatin Remodelling Agents

Reprogramming is reliant on chromatin remodelling, emphasising the need to improve the modification process. Although ooplasmic factors present in the oocyte can facilitate remodelling to some extent, they cannot modify all nuclei. One of the key events for successful remodelling is the unrestricted exchange of proteins between the ooplasm and the transferred nucleus that takes place at the DNA level [41]. Chromatin is tightly packed resulting in physical limitations to protein exchange; any approaches that would unravel this structure may be beneficial to the modification process [41]. Treatments that encourage the somatic genome to imitate normal DNA methylation and chromatin remodelling have been investigated to aid epigenetic reprogramming and ultimately cloning efficiency. These treatments entail preparing donor cells or treating SCNT embryos with a reagent that decreases DNA methylation, such as 5-aza-20-deoxycytidine, or a histone deacetylase inhibitor (HDACi) that increases histone acetylation. Improved SCNT embryo development and cloning efficiency have been described in several studies following use of only a few of these methods; however, no improvement has been reported following 5-aza-20-deoxycytidine treatment of donor cells before SCNT [18,112].

Reprogramming of somatic nuclei should occur before embryonic genome activation, therefore the unwinding of chromatin as a consequence of histone acetylation may effectively assist this process [41,113]. Global histone acetylation is achieved with HDACi, which functions by inhibiting histone deacetylases, causing chromatin structure alterations that allow proteins such as RNA polymerases to easily infiltrate the DNA and initiate transcription [16]. Histone acetylation in HDACi-treated SCNT embryos is increased and is beneficial after the reconstructed embryos have undergone activation [114]. Trichostatin A (TSA) is a commonly used HDACi in SCNT that enhances DNA demethylation [115]. In 2006, two independent groups established the appropriate conditions for TSA treatment of SCNT mouse embryos including concentration, time point, and period of exposure [15,116]. One group reported a fivefold increase in mouse clone survival by enhancing oocyte activation with TSA [15,117]. Blastocyst development rates in SCNT monkey embryos treated with TSA have improved from 4% to 18% [118]. Despite the enhanced blastocyst development rate, blastocyst quality and the possible establishment of stable ESCs remain unknown [118]. A recent study also confirmed improved blastocyst development and pregnancy rates using monkey SCNT embryos, after treatment with TSA at the one-cell stage [31]. In a different study, an assumption was made that high levels of TSA may have a negative impact on blastocyst quality, even though blastocyst formation was promoted with TSA treatment [35]. The most effective treatment protocol in mice is a TSA concentration between 5–50 nM, with constant exposure of the manipulated oocytes to TSA for at least 8–10 h from the start of activation, but before the first cell division [15,116]. Should TSA exposure of manipulated SCNT embryos exceed 12 h from the start of activation (specifically very close to the first cleavage stage), the embryos will arrest. Trichostatin A shows effectiveness from 5 nM but becomes toxic at 500 nM, hence the suggested TSA concentration of 5–50 nM [15]. Since the narrow window for successful reprogramming occurs before embryonic genome activation, the timing of TSA treatment is vital to its effectiveness on the developmental potential of the reconstructed embryos [113]. Based on the abovementioned guidelines for TSA treatment, protocols may differ in concentration and extent of TSA exposure. Although the rate of blastocyst formation is improved, another study reported that none of the SCNT embryos treated with TSA gave rise to animals that persisted through to adulthood [29]. In addition, TSA is teratogenic [119], therefore the use of high concentrations may be detrimental to the quality of blastocysts and will significantly reduce normal development of the embryo [16]. Scriptaid, another potent HDACi, has a lower toxicity level than TSA and, through its ability to increase transcription and protein expression, has resulted in significant improvements in the creation of cloned mice [16] and pigs [23,24]. Many studies have tried to use other HDACi to improve SCNT; however, TSA currently remains the best approach for mouse cloning, despite controversial outcomes in farm animals [120,121]. The mechanism through which HDACi improves the efficiency of cloning is most likely related to the capability to encourage the synthesis of nascent mRNA, following an increase in histone acetylation [16]. Even though incomplete, histone acetylation remodelling in reconstituted embryos is enhanced with HDACi treatment after SCNT [24].

## 6. Current Protocol Optimizations

One of the main purposes of creating SCNT blastocysts is isolation of the ICM to derive ESCs. However, poor-quality SCNT blastocysts will prevent ESC isolation [122]. Developmental delay can be seen in the SCNT group in Figure 6 below. Essentially, the defects in the SCNT group model the defects of failing human embryos, with characteristics such as poor-quality blastocysts, delayed development, fewer cells, and genome instability. Even though the rate of blastocyst development is encouraging, protocol optimization is continuously being undertaken to focus primarily on improving SCNT embryo quality. For example, monkey and human SCNT studies, supplementing spindle enucleation and the fusion medium with caffeine, have reported improved SCNT embryo development, blastocyst quality (characterized by visible and prominent ICM formation) and subsequent ESC line derivation [35,65]. In a study by Mallol and colleagues, the effect of supplementing the culture medium with vitamin C for at least 16 h after activation, in combination with Latrunculin A during micromanipulation and activation, significantly increased rates of mouse blastocyst formation [123]. The mean number of ICM cells at 96 h post activation and the total blastocyst cell number were also increased [123].

### Nuclear Reprogramming Strategies

Various factors can affect the success rate of blastocyst development in SCNT [61,62]. These include one or more of the many steps performed during SCNT, which may have negative effects on cytoplast quality, causing inefficient reprogramming and ultimately influencing the success of SCNT [35]. These factors may ultimately contribute to abnormal gene expression caused by epigenetic reprogramming failure of the donor somatic cell nucleus by the oocyte [33]. Continued research and protocol optimization, specifically nuclear reprogramming strategies, has been encouraged in order to improve the efficiency of nuclear reprogramming by SCNT [124] (Figure 7).

Histone deacetylase inhibitors including TSA and Scriptaid have been used successfully to improve the efficiency of SCNT blastocyst development in several species, including human, monkey, mouse, bovine and pig [12,15,23,31,35,118]. Nonetheless, the resulting blastocyst quality and the ability to establish stable ESCs remain unknown [118]. In addition, TSA toxicity [119] may be detrimental to the quality of the blastocyst, and may significantly reduce normal development of the embryo [16,35]. Based on gene expression analysis, Inoue and colleagues discovered the downregulation of X-linked genes (required for embryonic development) in SCNT mouse embryos; this is caused by the ectopic expression of the Xist gene that is responsible for X chromosome inactivation [125]. Birth rate and cloning efficiency in mice were dramatically improved by complete deletion of Xist from the donor genome or repression of the gene by injecting short interfering (si) RNA into SCNT embryos [125,126]. Furthermore, next-generation sequencing of donor cells and SCNT embryos identified histone 3 lysine 9 trimethylation (H3K9me3) of the somatic cell genome as the main obstacle preventing nuclear reprogramming [127]. A significant improvement in both mouse and human SCNT cloning competence was achieved by expressing H3K9me3 demethylase lysine demethylase 4 (Kdm4) D in SCNT embryos to reduce the H3K9me3 level [127,128,129,130]. The abovementioned studies suggest that by including nuclear reprogramming strategies into the SCNT procedure, an overall improvement in cloning can be achieved.

## 7. SCNT Applications

The importance of SCNT research in the mouse is highlighted by its distinct advantages over other experimental animal models and the potential to understand the underlying principles of nuclear reprogramming. The reversibility of these epigenetic processes facilitates many new prospects in basic research [70,124,131]. In humans, the idea of SCNT as a technique to generate specific ESCs from the somatic cells of an individual could ultimately lead to the understanding of disease mechanisms, as well as improve the efficiency of cell-based therapies for the treatment of degenerative diseases such as Parkinson’s disease with a negligible risk of immune rejection [4,131,132,133]. The integration of SCNT and ESC techniques refers to the injection of a patient’s somatic cell into an enucleated oocyte followed by the isolation of ESCs (which can differentiate into any cell type) from the cloned blastocysts. The differentiated cells are then grafted into the affected patient who donated the healthy somatic cell [124]. In 2012, the Nobel Prize for Physiology and Medicine was jointly awarded to Sir John Gurdon, the inventor of SCNT, and Shinya Yamanaka, the pioneer of induced pluripotent stem cell (iPSC) technology [124].

The potential to create new gametes for animals, and in future for human patients, was made more realistic by Hayashi and colleagues [134,135]. This group accomplished the production of viable sperm and oocytes from ESCs and iPSC derived germ cells. The gametes successfully completed development and produced several generations of offspring. Research comparing the ability of SCNT and iPSC derivations to produce gametes is of great interest and will continue in the future.

## 8. Ethical and Legal Implications of SCNT in Humans

Most countries and organisations have prohibited human reproductive cloning [136]. Human cloning in all forms has been banned by the United Nations, emphasising the incompatibility of human cloning with human dignity and the protection of human life [137]. Creating a human being that shares the same nuclear genome as another living or dead being is also prohibited by the European Council [138]. The International Society for Stem Cell Research recommends that the gestation or transfer of a human embryo created by SCNT or by other nuclear reprogramming techniques into a uterus should be banned [139]. Reproductive cloning is also prohibited in several countries that take part in large stem cell research programs, including the United States (US), China, Germany, and South Korea [136]. The use of reproductive SCNT as an option for the treatment of infertility has sparked widespread debate. The American Society for Reproductive Medicine (ASRM) has presented valid arguments against reproductive SCNT and concluded that it is unethical to use it as an assisted reproductive technology due to safety concerns, because of the developmental abnormalities seen in clones, and the undefined effect on children, relatives and humanity [140].

The technique of SCNT is presently performed in several laboratories worldwide to create human stem cells. In the United Kingdom (UK), human SCNT research is legal and in 2001 was included in the Human Fertilization and Embryology Act 1990 [141]. However, before performing SCNT, it is necessary to obtain permission from the Human Fertilization and Embryology Authority [142]. In the US, SCNT research is also legal but is not allowed to be funded by the national government because of the Dickey–Wicker Amendment bill passed in 1995 [32]. In the US, the Department of Health and Human Services and the National Institutes of Health forbid the use of funds for research studies involving the creation of human embryos and the destruction thereof [32]. Nonetheless, SCNT research aimed at producing human ESCs may be legally performed when funded by private or non-governmental organisations [143].

As reviewed by Cervera and Mitalipov [32], several ethical and legal implications are associated with SCNT. One of the main drawbacks of SCNT experimentation in humans is evidently the availability of oocytes. In addition to this limitation, there are financial and ethical implications related to obtaining human oocytes for research purposes. They argue that strict regulation of human SCNT research should be maintained by local institutional research ethics boards, as well as by ethical guidelines that have been established by the US National Academy of Science, the International Society for Stem Cell Research, and the ASRM [143]. Local laws regulating compensation for oocyte donors may also govern the procurement of human oocytes for research purposes. In California, for example, patients donating oocytes for research purposes are covered for certain expenses, but are not reimbursed for “time, effort and inconvenience” [143]. In Oregon, on the other hand, research oocyte donors are fully compensated in a manner that is equal to reproductive oocyte donors [144,145].

In the South African context, reproductive cloning is banned. Therapeutic cloning and research involving human oocytes and embryos, including SCNT, are all acceptable with the requirement that ministerial authorisation needs to be obtained [146]. However, the National Health Act does not address matters regarding oocyte donation specifically for research purposes. Regarding the payment of oocyte donors for reproductive oocyte donation in South Africa, details are provided in the 2008 guidelines of the Southern African Society for Reproductive Medicine and Gynaecological Endoscopy (SASREG) [147]. According to these guidelines, “monetary compensation of the donor should reflect the time, inconvenience, financial costs to the donor-e.g., travel, loss of income and childcare costs, physical and emotional demands and risks associated with oocyte donation and should be at a level that minimizes the possibility of undue inducement of donors and the suggestion that payment is for the oocytes themselves. The monetary compensation should not be predicated on the clinical outcome (no. of oocytes or pregnancy outcome) but rather on fair compensation for the procedure of donating eggs. Donors should only receive financial compensation via fertility clinics and not receive any compensation directly from the recipients or other third parties”. An amendment was made to the SASREG Guidelines for Gamete Donation on the 25 November 2014, which now states that “egg donors should not be compensated more than R 7000.00 per procedure from 1 January 2015” [148].

Alternative sources of human oocytes have been investigated for SCNT research because of the financial and ethical burdens related to reimbursement of oocyte donors. Immature oocytes, which are generally discarded in assisted reproductive procedures, are voluntarily donated by patients for research purposes [32]. However, in vitro maturation, fertilization, and subsequent development of these immature oocytes to the blastocyst stage are highly compromised following SCNT and they are therefore not appropriate for optimization of the SCNT procedure [149]. An ideal approach would be to collect donors willing to donate oocytes solely for research purposes without reimbursement; however, most women are simply not prepared to undergo ovarian stimulation and invasive oocyte retrieval without being reimbursed for their efforts [150].

It should be noted that even the use of high-quality human oocytes in SCNT does not guarantee successful embryo development to the blastocyst stage [122]. In a study conducted by Noggle and colleagues [122], prior to the use of an HDACi and a different activation protocol, successful blastocyst development and subsequent isolation of ESCs were only observed in those embryos that had somatic cells transferred to non-enucleated oocytes. This observation may imply that unknown factors essential for effective reprogramming of the somatic cell nucleus may be removed during enucleation and are retained by the presence of the oocyte’s own meiotic spindle complex. Another study reporting early failure in monkey SCNT embryo development also assumed the cause was related to the removal of reprogramming factors during enucleation [1]. Subsequent research has, however, shown that the oocyte meiotic spindle complex is not a prerequisite for the successful reprogramming of the somatic cell nuclear genome, especially when an HDACi such as TSA is used [60].

Nevertheless, many studies encourage that each step in SCNT be thoroughly optimized and adapted specifically for human oocytes. However, a major drawback related to rigorous testing on human oocytes and also the main obstacle for therapeutic cloning is the requirement for many good-quality human oocytes, which remains limited [32].

## 9. Conclusions

From the theoretical and epigenetic perspective, mouse SCNT has provided important information for overall technical improvements of SCNT techniques [124]. However, the main outcome from SCNT studies is that we understand the differentiated state much better. The cytoplast plays an important role in SCNT by preventing premature activation. Additionally, careful fusion of the somatic cell and efficient activation are required to effectively reprogram the somatic cell nucleus to the pluripotent state [131]. The quality of oocytes has always been important and appreciated, where positive SCNT outcomes have correlated with excellent donor oocytes in both non-human primates [60] and humans [35,37]. Soon, oocyte quality may be considered a less detrimental factor as continued protocol modifications are made [131]. The derivation of human ESCs has been achieved at high rates from SCNT-produced blastocysts, which is critical to the potential use of SCNT as a method for therapeutic cloning [35,37,129]. Technical advances in the SCNT protocol include polarized light imaging for the removal of the meiotic spindle complex from the oocyte, incubation of the donor somatic cell with a membrane fusogen (e.g., HVJ), and laser-assisted nuclear transfer to facilitate donor cell insertion into the PVS and improvements to human oocyte activation [31,124]. However, despite these advances, poor success rates have been observed in terms of SCNT embryo development, which may be attributed to the incomplete or inefficient reprogramming of the somatic cell nucleus required to support ongoing development [91,104,131,151]. An important reason for creating SCNT blastocysts is to derive ESCs following isolating the ICM. The derivation of human ESCs has been achieved from laser-assisted as well as piezoelectric SCNT-produced blastocysts [35,37,129]. The required blastocyst quality expected for efficient ESC derivation after SCNT has not been specified in the literature. However, poor-quality SCNT blastocysts will prevent ESC isolation [35,122]. Continued research and protocol optimization in terms of nuclear reprogramming strategies should be encouraged for improvement of reprogramming efficiency by SCNT.

### 9.1. Recommendations

Establishing this protocol requires a significant institutional commitment and a dedicated research program with researchers skilled at manipulating oocytes and embryos [36]. The technique is constantly developing, and based on current research and literature the following recommendations for an optimized SCNT procedure using a mouse model can be made:

#### 9.1.1. Equipment and Oocyte Handling

The piezoelectric micromanipulation system may be used for single step nuclear transfer to improve time management and avoid the purchase of a laser system and a membrane fusogen kit [9,61,70].The optimal donor oocytes to use for high developmental competence, minimal susceptibility to in vitro handling and micromanipulation and visibility of the meiotic spindle complex, are 8–12-week-old B6D2F1 mice [61].Undesirable fluctuations in the culture environment of oocytes through exposure outside of the incubator for longer than 20–30 min can be prevented by adjusting the number of oocytes undergoing experimentation to the level of experience of the operator [70].If the meiotic spindle complex cannot be visualized at first, it can be identified using Hoechst and UV illumination, which would be for practice purposes only [70].

#### 9.1.2. Medium Supplementation

Supplementing the culture medium with an antioxidant such as vitamin C (that protects cells against reactive oxygen species) for at least 16 h after activation should be considered, which in combination with Lat-A during micromanipulation and activation may increase the rates of blastocyst formation [123].Due to its lower toxicity, Scriptaid may be added to the activation protocol and culture medium as an alternative histone deacetylase inhibitor to TSA, to promote histone acetylation and ultimately improve SCNT embryo development [16,23,24].

#### 9.1.3. Quality Control and Training

It is important to create laboratory-based standard operating procedures and training manuals for establishing and optimizing methods of SCNT.Maintaining a stable environment for embryo development including temperature, pH and medium optimization should never be underestimated and is of utmost importance.Non-enucleated oocytes should be activated and cultured in parallel with reconstructed oocytes as parthenogenetic controls for the activation protocol and culture conditions [70].To ensure optimal use of this advanced technique, adequate training of embryologists in intracytoplasmic sperm injection and biopsy procedures is a prerequisite [61,62].Many hours of experimentation and practice are required to perform the complex SCNT protocol [61,62].

Despite the many technical, ethical, and legal implications associated with SCNT, this field of research holds great potential. In addition to the extensive practical applications of SCNT, the technique may offer unique and intriguing experimental approaches to genomic research, and more specifically to epigenetics [92]. Research in this field may provide insight into the reprogramming ability of the somatic cell genome to a totipotent state, identical to that of a fertilized oocyte [120]. The importance of SCNT research in the mouse is emphasised and encouraged by the potential to understand underlying principles of nuclear reprogramming. The reversibility of these epigenetic processes will facilitate the emergence of new prospects in basic research, and almost certainly, in time to come, cell transplantation and regenerative medicine [70,124,131].

## Figures and Tables

**Figure 1 ijms-21-02314-f001:**
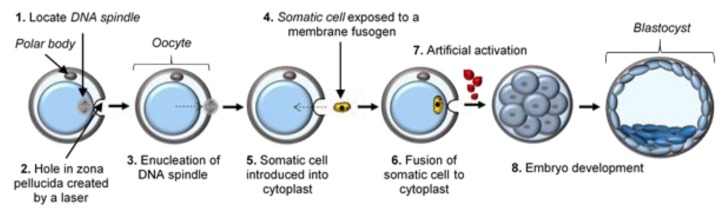
Schematic representation of the steps (1–8) involved in laser-assisted SCNT. Dashed arrows represent removal of the spindle in step 3, and transfer of the somatic cell into the cytoplast in step 5.

**Figure 2 ijms-21-02314-f002:**
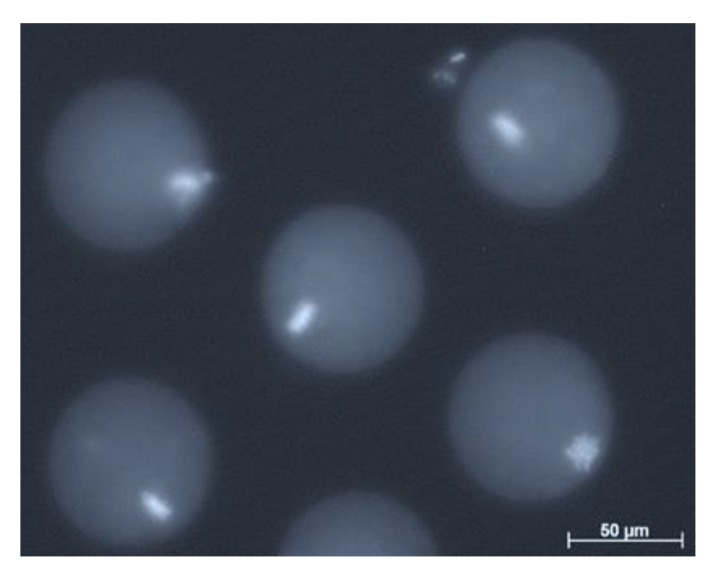
Fluorescent microscopy image of Hoechst 33342 DNA stained B6D2F1 mouse oocytes (image captured by C.G. as part of her Masters dissertation at the Reproductive Biology Laboratory, University of Pretoria).

**Figure 3 ijms-21-02314-f003:**
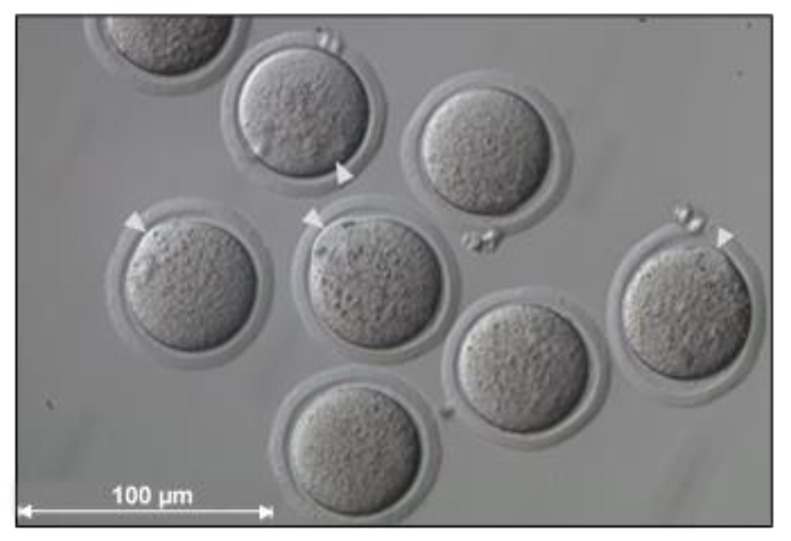
Photo image of B6D2F1 mouse oocytes. Meiotic spindle complexes (visible as a translucent region with an occasional bulge) are indicated by white arrowheads (image captured by C.G. as part of her Masters dissertation at the Reproductive Biology Laboratory, University of Pretoria).

**Figure 4 ijms-21-02314-f004:**
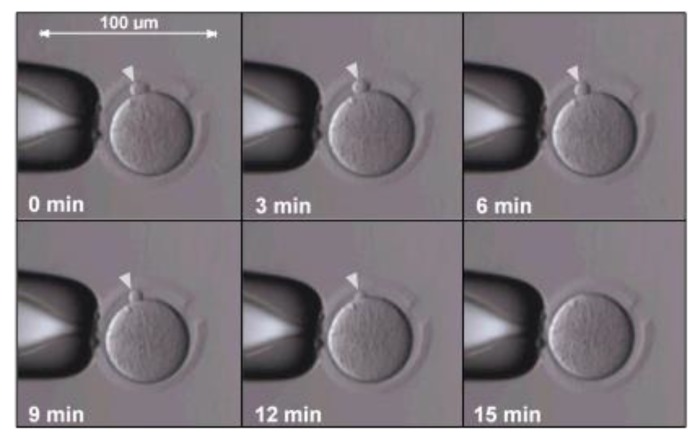
Time-lapse images showing fusion of a cumulus cell into a cytoplast within 15 min. The transferred cumulus cell is indicated by white arrowheads (images captured by C.G. as part of her Masters dissertation at the Reproductive Biology Laboratory, University of Pretoria).

**Figure 5 ijms-21-02314-f005:**
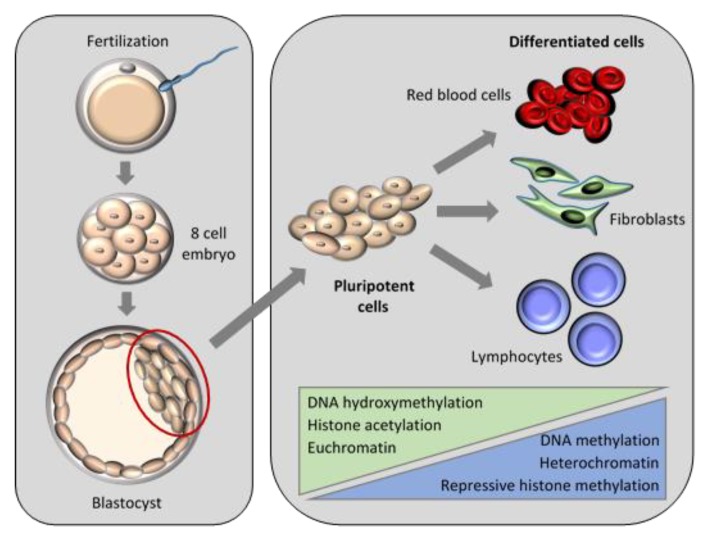
Epigenetic interactions during the progression from pluripotent to differentiated cells.

**Figure 6 ijms-21-02314-f006:**
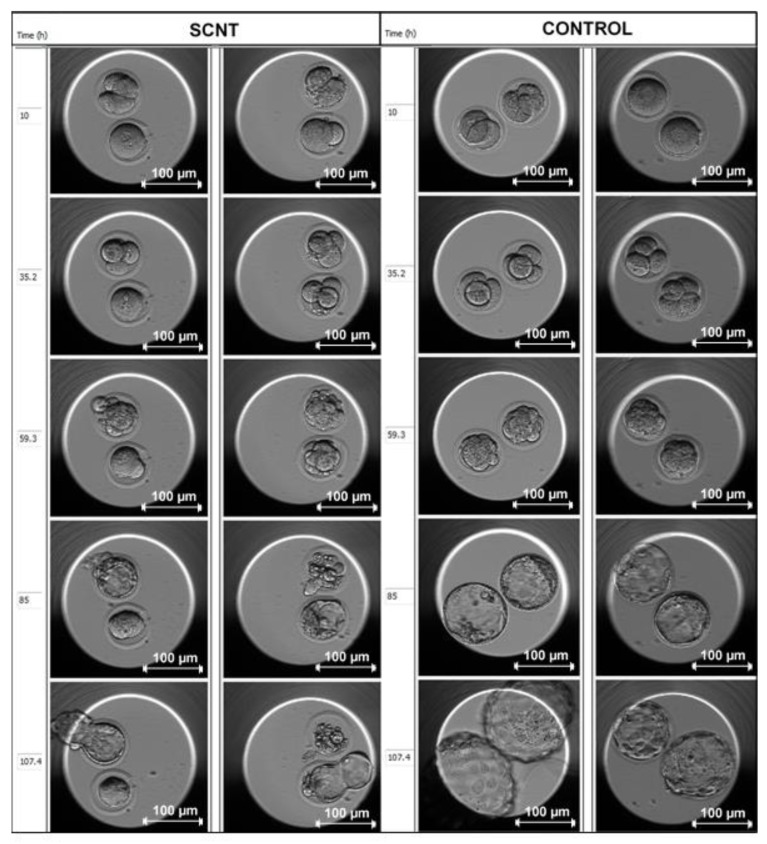
EmbryoScope™ time-lapse system images comparing the development of poor-quality SCNT BDF1 mouse embryos versus non-enucleated, parthenogenetic control BDF1 mouse embryos (images captured by C.G. as part of her Masters dissertation at the Reproductive Biology Laboratory, University of Pretoria).

**Figure 7 ijms-21-02314-f007:**
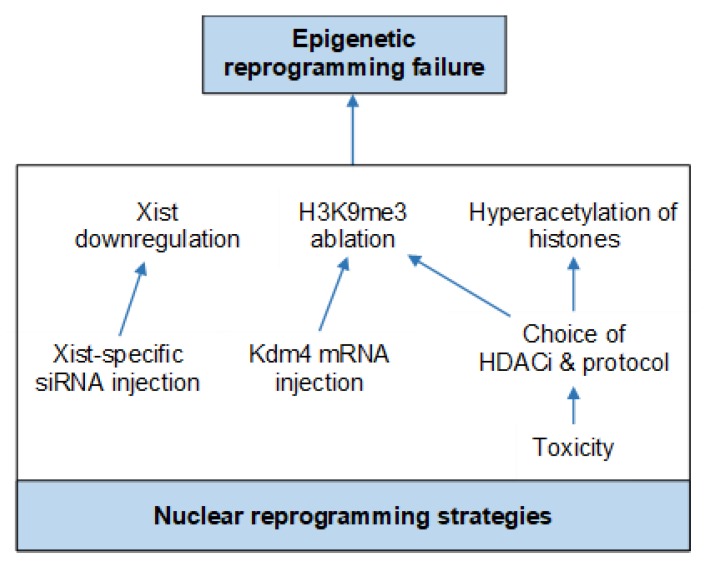
Diagram illustrating the most recent and applicable nuclear reprogramming approaches developed to improve epigenetic reprogramming failure.

**Table 1 ijms-21-02314-t001:** The success rate of full-term development of mice cloned from different donor cells.

Donor Cell Type	Donor Age	Success Rate	References
Cumulus (B6D2F1)	Adult	2.5–4.5%	[9,82]
Cumulus (129B6F1)	Adult	3.2%	[83]
Cumulus (BDF1x129/Sv)	Adult	15.6%	[84]
Tail-tip fibroblast	Adult	1.1–4.8%	[51,85]
Fetal fibroblast	Fetus	3.0–3.7%	[71,82]
Sertoli (B6D2F1)	Newborn	4.5%	[72]
Sertoli (B6129F1)	Newborn	10.8%	[83]
Neuronal stem cell	Newborn	0.5%	[74]
Neuronal stem cell	Fetus	1.6%	[76]
Hematopoietic stem cell	Adult	0.7%	[75]
Keratinocyte stem cell	Adult	5.4%	[77]
ESC (G1 phase)	Embryonic	12.3%	[57]
ESC (G2/M phase)	Embryonic	6.4%	[57]

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
