# Peer review of "Lessons Learned from Somatic Cell Nuclear Transfer"

_ijms, 2020, doi:10.3390/ijms21072314_

Round 1
Reviewer 1 Report
This review manuscript was well summarized the recent progress of mammalian SCNT technology and suggest the way to go for SCNT study. Only minor issues should be corrected for publication.
- Line 555 of page 15: all results were not acquired from laser-assisted SCNT-produced blastocysts. At least, one group didn't use laser system. They usually used Piezo system for zona penetration.
- In all manuscripts, "sertoli cells" should be changed to "Seroli cells"
- In Fig. 7: Kdm4d or Kdm4a, or Kdm4 is better.
Reviewer 2 Report
Dear Authors,
this is a nice review and I think it is vital for the field of embryogenesis. after reading the content, I think that the way this review is written requires re-submission due to the vagueness of the message o that the authors aim to send.
the abstract has a general theme without being focused on what the authors intend to discuss in their review. In fact, the abstract doesn't clarify if this is a review or a report as well it doesn't cover the general and specific themes that the review is tackling.
I would request that authors avoid general sentences that require highly specialized readers to grasp what is means such as the first sentence written: Biologically, somatic cell nuclear transfer (SCNT) reverses the differentiated state.
The same applies to the sections in the review (is this a review, if so please indicate in the abstract), they need to be connected to each other and they need to form a synthesized whole connected section.
The conclusion should also indicate what are the recommendations to the readers.
Round 2
Reviewer 2 Report
Thank you, good work